# Comparative Coexpression Analysis of Indole Synthase and Tryptophan Synthase A Reveals the Independent Production of Auxin via the Cytosolic Free Indole

**DOI:** 10.3390/plants12081687

**Published:** 2023-04-18

**Authors:** Yousef M. Abu-Zaitoon, Ezz Al-Dein Muhammed Al-Ramamneh, Abdel Rahman Al Tawaha, Sulaiman M. Alnaimat, Fouad A. Almomani

**Affiliations:** 1Department of Biology, Faculty of science, Al-Hussein Bin Talal University, Maan 71111, Jordan; abdeltawaha@yahoo.com (A.R.A.T.);; 2Department of Agricultural Sciences, AL-Shouback University College, Al-Balqa Applied University, Maan 71911, Jordan; 3Department of Applied Biology, Jordan University of Science and Technology, Irbid 22110, Jordan

**Keywords:** indole synthase, tryptophan synthase A, tryptophan synthase B, coexpression, logit score

## Abstract

Indole synthase (INS), a homologous cytosolic enzyme of the plastidal tryptophan synthase A (TSA), has been reported as the first enzyme in the tryptophan-independent pathway of auxin synthesis. This suggestion was challenged as INS or its free indole product may interact with tryptophan synthase B (TSB) and, therefore, with the tryptophan-dependent pathway. Thus, the main aim of this research was to find out whether INS is involved in the tryptophan-dependent or independent pathway. The gene coexpression approach is widely recognized as an efficient tool to uncover functionally related genes. Coexpression data presented here were supported by both RNAseq and microarray platforms and, hence, considered reliable. Coexpression meta-analyses of Arabidopsis genome was implemented to compare between the coexpression of *TSA* and *INS* with all genes involved in the production of tryptophan via the chorismate pathway. Tryptophan synthase A was found to be coexpressed strongly with *TSB1/2*, *anthranilate synthase A1/B1*, *phosphoribosyl anthranilate transferase1*, as well as *indole-3-glycerol phosphate synthase1*. However, *INS* was not found to be coexpressed with any target genes suggesting that it may exclusively and independently be involved in the tryptophan-independent pathway. Additionally, annotation of examined genes as ubiquitous or differentially expressed were described and subunits-encoded genes available for the assembly of tryptophan and anthranilate synthase complex were suggested. The most probable TSB subunits expected to interact with TSA is TSB1 then TSB2. Whereas TSB3 is only used under limited hormone conditions to assemble tryptophan synthase complex, putative TSB4 is not expected to be involved in the plastidial synthesis of tryptophan in Arabidopsis.

## 1. Introduction

Indole-3-acetic acid (IAA), the dominant naturally occurring auxin phytohormone, affects a wide array of developmental and growth processes in plants [1,2,3]. Auxin controls cell division, cell expansion, differentiation of vascular tissue, initiation of adventitious and lateral root initiation, phototropism and geotropism, maintenance of apical dominance, fruit ripening, fruit and leaf abscission, as well as delay of leaf senescence. Even though it is the most important hormone, auxin production via only one route has been fully elucidated in plants comparing to two pathways in bacteria. IAA is synthesized from both tryptophan (trp) and trp precursors via trp-dependent pathways and a trp-independent one, respectively [4,5,6]. Several intermediates including indole-3-acetamide, indole-3-acetaldoxime, indole-3-pyruvic acid, and tryptamine were reported to produce IAA from trp [5,7,8,9,10]. The indole-3-pyruvic acid pathway, represented by the concerted action of tryptophan aminotransferase and YUCCA, is the solely dissected route reported to predominate IAA synthesis in plants [11,12]. In this pathway, tryptophan amino transferase and tryptophan aminotransferase related enzymes converts tryptophan into indole-3-pyruvic acid, whereas YUCCA, a flavin monooxygenase enzyme, catalyze the conversion of indole-3-pyruvic acid into indole-3-acetic acid.

The plastidial synthesis of trp from chorismate via the chorismic acid pathway has long been elucidated basically by implementing forward genetic approaches [13,14]. In this pathway, chorismate is initially converted to anthranilate via anthranilate synthase (AS). Anthranilate is then converted to phosphoribosyl anthranilate via phosphoribosyl anthranilate transferase (TRP). After that, phosphoribosyl anthranilate isomerase (PAI) converts phosphoribosyl anthranilate into 1-(o-carboxyphenylamino)-1-deoxy-ribulose 5-phosphate which is then converted into indole-3-glycerol phosphate by indole-3-glycerol phosphate synthase (IGPS). Conversion of indole-3-glycerol phosphate to indole and subsequently to trp is catalyzed by the α and β subunits of tryptophan synthase (TS; Figure 1).

The importance of searching for other IAA synthesizing pathways comes from the observations that double, triple, and quadruple knockout of genes encoded the indole-3-pyruvic acid pathway enzymes have not found to completely block IAA synthesis, suggesting a role for other enzymes in IAA production in plants. Indole or indole-3-glycerol phosphate is the potential branch point reported to synthesize IAA via a trp-independent pathway [15]. However, this pathway is largely unknown in terms of enzymes, and intermediates [16,17]. Recently, indole synthase (INS), a homologous cytosolic enzyme of the plastidal tryptophan synthase A, has been reported to convert indole-3-glycerol phosphate to indole in Brassicaceae including *Arabidopsis thaliana* and *Polygonum tinctorium* [18,19]. A pivotal role for the INS pathway in embryo development was suggested. In *Zea mays*, *BX1*, and *IGL* genes encode cytosolic indole synthase responsible for the production of free indole were investigated [20]. The importance of indole as a precursor for the production of IAA in the trp-independent pathway was further highlighted by the use of yucasin DF (YDF), a competitive inhibitor of yucca enzyme, increase labelling of IAA from indole, and to less extent from anthranilate [21,22]. Moreover, a suppressor mutant of tryptophan aminotransferase, *iss*, grown on indole exhibited an eightfold higher level of IAA compared to wild type [23]. In a previous study and based on coexpression analysis, amidase was suggested to interact with INS in *Arabidops* is via the trp-independent pathway of IAA synthesis [24].

A limited role for the INS pathway in IAA synthesis was suggested in the rosette of Arabidopsis based on the observation that the transcript level of *INS* is seven times lower than that of *TSA* [17]. However, in *Polygonum tinctorium*, *INS* transcript is five times higher that of *TSA* [19]. Nonhebel [25] expected that the cytosolic INS occurs only in the member of Brassicaceae and Cleomaceae closest plant family species and, therefore, a restricted role for this enzyme in IAA synthesis is suggested. The importance of INS was further challenged by a proposal that it may form a complex with the plastidal localized TSB4 or its free indole may be passed to TSB4, converted to trp, and then to IAA via the trp-dependent pathways [25].

Analysis of gene function by loss-of-function is an expensive and time-consuming approach. Whereas function of only 24% genes of the model dicot plant Arabidopsis has been mapped, only 1% of important crops genes including maize and rice has been experimentally characterized [26]. Gene coexpression approach is widely recognized as an efficient tool to uncover functionally related genes [27,28,29,30]. In this piece of work, the involvement of INS in either trp-dependent or independent pathways was examined by the implementing of coexpression meta-analyses of Arabidopsis genomic data. Coexpression of *INS* with all genes-encoded enzymes in the chorismic acid pathway was compared to that of *TSA.* A potential involvement of INS in the trp-dependent pathway could be hypothesized if INS is coexpressed with one or more genes reported to encode enzymes in the plastidial synthesis of trp. Tryptophan synthase A, an INS homologue, that is well known to act in the trp-dependent pathway was used as a control. To achieve this goal, ATTED-II, the Arabidopsis coexpression database supported by the large number of samples as well as the high reliable microarray and RNA sequencing data, was implemented.

## 2. Methods

The coexpression data obtained from the ATTED-II database has been used in numerous studies to identify potential functions of uncharacterized genes in *Arabidopsis thaliana*. For instance, a recent study utilized the coexpression network approach to identify genes involved in the regulation of flowering time in *Arabidopsis thaliana* [31]. The authors identified a novel gene, At5g23430, which was significantly coexpressed with multiple known flowering time regulators, and demonstrated its role in the regulation of flowering time. In this study, version 11 of ATTED-II database released in 2021 was used to extract coexpression information for the target genes of *Arabidopsis thaliana*. Coexpression data obtained from this database is supported by the two platforms: RNA sequencing (RNAseq; Ath-r.c5-0) and microarray platforms (Ath-m.c9-0), as well as the unified or default coexpression platform (Ath-u.2). The coexpression analysis tool (CoExSearch; Coexpressed gene list from multiple query genes) available on the ATTED-II (http://atted.jp (accessed on 28 February 2023)) database was used to find genes significantly coexpressed with the *Arabidopsis INS* and *TSA* genes. The available top 2000-gene lists positively and 300-gene lists negatively coexpressed with the searched seed genes were studied and analyzed. Gene lists significantly coexpressed with the target genes were obtained from both specific samples in certain conditions including: hormone, tissue, biotic, and abiotic stress, as well as in general conditions (condition-independent manner). All genes previously reported to have an important role in indole and trp synthesis in the plastid and cytosol were analyzed in this work (Table 1).

The Logit Score (LG) is an improved coexpression index provided as a z score. It is a powerful coexpression index that was utilized in this study for its improved accuracy and precision [32]. This is because it takes into account multiple factors such as the degree of coexpression, the specificity of the coexpression, and the statistical significance of the coexpression. The significance of this score was compared to the mutual rank score used in the previous release. Several pathways with confirmed enzymes catalyzing reactions show that LG score is more valuable and powerful. New coexpression ranks for all genes strongly coexpressed with *INS* and *TSA* as a bait set were investigated as well. This approach was also attempted via using *TSA*, *INS*, as well as the four *TSB* homologues as a bait set to find out the possibility of association of their products as subunits in tryptophan synthase complex [33]. The same approach was used to identify genes that mainly participate in the assembly of anthranilate synthase complex.

## 3. Results and Discussion

### 3.1. Tryptophan Synthase A but Not Indole Synthase Is Coexpressed with Tryptophan Synthase B Genes

Indole-3-acetic acid is synthesized from a trp via trp-dependent pathways or upstream trp precursors via a trp-independent one. Indole synthase, converts indole-3-glycerol phosphate into indole, is the first enzyme proved to act in the trp-independent pathway in the cytoplasm of *Arabidopsis thaliana* [18]. Arabidopsis was found to contain only one gene encodes *INS* (At4g02610). Alternatively, tryptophan synthase α subunit encoded by *TSA* produces indole from indole-3-glycerol phosphate in the plastids [34,35]. After that, TSβ subunit encoded by *TSB* converts indole into trp. Like *INS*, Arabidopsis possess one experimentally proven *TSA* gene (At3g54640) and two *TSB* genes (*TSB1*; At5g54810, *TSB2*; At4g27070). Now, proteome data reveals another two homologues of *TSB* genes (*TSB3*; At5g38530, *TSB4*; At5g28237). The importance of INS in the trp-independent pathway was questioned as INS was suggested to interact with TSB to produce trp and then to IAA via trp-dependent pathway [25]. This suggestion was investigated through the analysis of coexpression data of Arabidopsis genome available on the ATTED-II database [32].

Analysis of coexpression data supported by RNA sequencing and microarray platforms under general and specific conditions clearly shows that *INS* is not coexpressed with *TSB1*, *TSB2*, *TSB3*, or *TSB4*, suggesting that INS is not involved in trp production. Our data contradicts Nonhebel suggestion [25], who expected the availability of TSB4 to convert free indole produced by INS to trp. Therefore, INS is expected to produce indole from indole-3-glycerol phosphate for IAA synthesis in the alternative cytoplasmic pathway. Our data is in agreement with Ouyang et al., [16] who suggested two independent pathways for indole synthesis: one via INS to produce IAA and many other secondary metabolites, whereas the other indolic pool is synthesized via tryptophan synthase and used to synthesize IAA via the tryptophan dependent pathway. The observation that INS is not coexpressed with the TSB genes suggests that it has a distinct role in the biosynthesis of indole and IAA. Our data provide evidence for the existence of two independent pathways for indole synthesis, which may have different regulatory mechanisms and physiological functions. Moreover, the identification of INS as a key enzyme in the alternative cytoplasmic pathway for IAA synthesis could have important implications for plant growth and development. Further studies are needed to elucidate the precise mechanisms of the regulation of indole and trp biosynthesis in plants and to explore the potential applications of these pathways for improving plant growth and development under different environmental conditions.

On the other hand, *TSA* is significantly coexpressed with *TSB1* and *TSB2* in the Ath-u.2, and Ath-r.5 platforms under general conditions. The significant coexpression between *TSA* and *TSB3* were found only under hormone-specific condition, whereas no coexpression was observed with *TSB4* (Table 2). The significant coexpression between *TSA* and *TSB1/B2* genes led us to find the new coexpression index for the three bait genes. The top 300 genes list significantly coexpressed with the bait set was not found to contain *INS*, or even *TSB3/B4.* Tryptophan synthase is a heterotetrameric complex of TSA and TSB [13]. In this piece of work, the higher coexpression relation between *TSA* and *TSB1* (Table 2) clearly suggests that TSB1 is the preferred subunit for the assembly of the tryptophan synthase complex. This result is in agreement with the involvement of TSB1 in producing 90% of tryptophan in leaves of Arabidopsis under standard growth conditions [34]. The importance of *Tryptophan synthase B2* as the next choice for the assembly of the tryptophan synthase complex is also in agreement with a report that showed the involvement of TSB2 in producing 10% of the tryptophan without being subjected to organ-specific variation [36]. Moreover, the expression of *TSB2* was noticed only under specific low-light growth conditions [34].

The availability of TSB3 to interact with TSA only under a specific hormone condition may be explained by its ability to convert indole and seine to tryptophan [37]. It has been suggested that TSB3 indirectly interacts with TSA. Correlation at the translation level may clarify the importance of this gene product in trp biosynthesis. *Tryptophan synthase B4* was not found to coexpressed with *TSA* in any examined platforms or conditions. Even though TSB4 was reported to be conserved throughout the plant kingdom [38], no significant direct role for trp production is expected for this gene product, at least in Arabidopsis. Another explanation for this finding is the involvement of the TSB4 gene product in trp synthesis independent of the TSA subunit [37].

The cytosolic volatile indole in maize was reported to be produced apart from the action of tryptophan synthase A [39]. In this article we suggest that, in Arabidopsis the free indole produced by the action of indole synthase is not converted into tryptophan by tryptophan synthase B1, B2, B3, or tryptophan synthase B4 and is, therefore, not involved in the tryptophan-dependent pathway of IAA synthesis.

### 3.2. Indole Synthase Is not Coexpressed with the Upstream Tryptophan Synthase Genes in the Chorismic Acid Pathway

Coexpression between *INS* and the remaining genes known to be involved in the trp biosynthesis via the chorismic acid pathway other than tryptophan synthase genes was further analyzed and compared with that of *TSA*. The anthranilates synthase α subunits encoded by anthranilate synthase A gene (*ASA1*; At5g05730) catalyzes the committed reaction in trp synthesis by converting chorismate into anthranilate. The arabidopsis genome contains two paralogs of *ASA1* (*ASA2*; At2g29690, *ASA3*; At3g55870) [40]. Anthranilate synthase β subunit which constitutes a heterotetramer complex with ASA is encoded by *ASB1*; At1g25220 and *ASB2*; At5g57890. Additionally, Arabidopsis genome was reported to possess four putative *ASB* genes including: At1g25155, At1g25083, At1g24909, and At1g24807. Sequences of the last four genes are very similar and appear as one cluster on chromosome 1 [41,42]. Analysis of coexpression between *INS* and anthranilate synthase genes unambiguously revealed that *INS* is not coexpressed with these genes in the general or specific conditions in all examined platforms.

On the other hand, *tryptophan synthase A* is significantly and strongly coexpressed with *ASA1* under general and specific conditions in all examined platforms (Table 3). The remaining *ASA* isoforms were not coexpressed with *TSA*. Additionally, *ASB1* is the only isoform of *ASB* that coexpressed with *TSA* in the Ath-u.2 and Ath-r.5 platforms. When the three isoforms of *ASA* and five isoforms of *ASB* were coexpressed as a bait set, only *ASA1* and *ASB1* were coexpressed significantly with each other suggesting that these two gene products are the major components involved in the formation of tetramer complex. The essential importance of ASA1 and ASB1 was documented in several physiological activities including the formation of adventurous root [43]. Even though both subunits of anthranilate synthase are involved in the assembly of heterotetrameric complex, anthranilate could be produced by the activity of α-subunits alone. The essential importance of *ASA* compared to *ASB* is unambiguously clear in this study as *TSA* were found to be coexpressed with *TSA* under all examined conditions. Additionally, data presented here may indicate that *ASA1* is constitutively expressed in Arabidopsis under general growth conditions, whereas the expression of *ASB1* may be induced by elicitation. The same conclusion was drawn by [44]. However, the remaining isoforms of ASA and ASB could be also involved in the production of anthranilate as a double *ASA1/ASB1* mutant; this had no effects on trp or auxin phenotypes [41].

The second step in trp biosynthesis is the conversion of anthranilate to phosphoribosyl anthranilate via phosphoribosyl anthranilate transferase (*TRP1*; At5g17990, *TRP2*; At1g70570). Analysis of microarray data revealed that unlike *INS*, *TSA* is strongly coexpressed with phosphoribosyl anthranilate transferase1 (*TRP1*) in all platforms, under general and specific conditions examined in this work. This result suggests that *TRP1* is the main phosphoribosyl anthranilate transferase isoform involved in trp biosynthesis in Arabidopsis. After that, the conversion of phosphoribosyl anthranilate to 1-(o-carboxyphenylamino)-1-deoxy-ribulose 5-phosphate is catalyzed by phosphoribosyl anthranilate isomerase. Three isoforms of *PAI* genes are found in Arabidopsis including *PAI1*; At1g07780, *PAI2*; At5g05590 as well as *PAI3*; At1g29410 [45]. Analysis of coexpression data revealed that neither *TSA* nor *INS* coexpressed with any of the *PAI* genes in any platforms or conditions available on the ATTED-II database. The possible explanation for this result may be the central importance of this gene product in pathways other than trp biosynthesis. Alternatively, a possible role for this enzyme in indole production under specific hormone or stress conditions is not excluded.

The fourth enzyme in the trp biosynthesis pathway is indole-3-glycerol phosphate synthase that produces indole-3-glycerol phosphate from 1-(O-carboxyphenylamino)-1-deoxyribulose-5-phosphate. Arabidopsis was reported to have two functional *indole glycerol phosphate synthase* genes: *IGPS1* (At2g04400) as well as *IGPS2* (At5g48220) [46]. Analysis of coexpression between *INS* and *IGPS* genes unambiguously revealed that *INS* is not coexpressed with these genes. Even though both enzymes were confirmed to produce IGP [17], only *IGPS1* is significantly and strongly coexpressed with *TSA.* The first gene among 21,500 genes available in the Arabidopsis genome found to be strongly coexpressed with *TSA* is *IGPS1*, suggesting that it is the predominant *IGPS* isoform in the trp biosynthesis in Arabidopsis (Table 3). The pivotal importance of IGPS1 in trp biosynthesis is in agreement with the gene-knockout research conducted in maize. Arabidopsis IGPS1 was found in the same clade as IGPS2 maize homologue strongly interacted with TSA [15]. On the other hand, further experimental analysis of expression patterns of the Arabidopsis IGPS2 gene may explain its specific function in indole homeostasis. When the 16 isoforms of *ASA*, *ASB*, *TRP*, *PAI*, and *IGPS* were coexpressed as a bait set only *ASA1*, *ASB1*, *TRP1*, and *IGPS1* were significantly coexpressed with each other. Additionally, *TSA1* and *TSB1*/*B2*, but not *INS*, were found to be significantly coexpressed with the bait set query genes in the top 300 genes list (Table 4).

To sum up, among all genes-encoded enzymes in the plastidial synthesis pathway of indole, no genes were found to be coexpressed with the cytosolic indole synthase. Whereas, the plastidial *INS* homologue, *TSA*, was found to be strongly and significantly coexpressed with *anthranilate synthase A1*, *anthranilate synthase B1*, *phosphoribosyl anthranilate transferase1*, as well as *indole glycerol phosphate synthase1* genes. On the other hand, neither *INS* nor *TSA* were found to be coexpressed with any isoform of the *Phosphoribosyl anthranilate isomerase genes.*

## 4. Conclusions

Indole has long been elucidated as an important substrate of IAA production in plants. Intermediates and enzymes in the indole synthesis pathway is well established in plastids. In this pathway, indole synthesized by the action of tryptophan synthase α subunit is not released but directly converted to tryptophan by the tryptophan synthase β subunit. However, indole synthase, the first enzyme proven to act in the cytosolic production of free indole, has recently been uncovered in Arabidopsis. In addition to feeding and measurement analysis, examination of loss of function *ins* mutants clearly showed the dramatically impaired trp-independent pathway of IAA synthesis during early embryogenesis. The importance of INS as a crucial enzyme in the trp-independent pathway was questioned mainly through bioinformatics analysis. Indole produced by INS was expected to be converted by TSB to produce trp after translocating to plastids and, hence, involved in the trp-dependent pathway. Therefore, the major goal of this work was to find out evidence for or against the involvement of INS in the trp-dependent pathway of IAA synthesis.

Gene coexpression analysis implemented in this work has long been effectively used to identify many pathways’ constituent genes. Analysis of coexpression data for the Arabidopsis genome, deposited on the ATTEDII database, unambiguously shows that no coexpression was found between the gene of INS and any enzyme-encoded genes involved in the production of indole or trp in the plastids. On the other hand, the importance of the plastidial indole synthase homologues, TSA, in the tryptophan-dependent pathway was confirmed by coexpression analysis as it was found to be positively strongly coexpressed with most enzyme-encoded genes in the chorismic acid pathway including anthranilate synthase A1/B1, phosphoribosyl anthranilate transferase1, indole-3-glycerol phosphate synthase1, as well as tryptophan synthase B1/2. Together these data suggest that indole synthase may be independently and separately involved in the trp-independent pathway of IAA synthesis. Like other aromatic amino acids, a complete or partial cytosolic pathway producing indole or trp is not excluded.

The reliability of data was confirmed as the same result was found true for using Arabidopsis data obtained from both general and specific experimental conditions including: hormone, tissue, biotic, and abiotic. Additionally, samples deposited on the ATTEDII database were collected from more than 21,000 genes and supported by RNAseq and microarray platforms. Moreover, prediction of gene function based on coexpression analysis was found to be consistent with the experimentally confirmed results on the available subunits for the assembly of anthranilate, and tryptophan synthase complex. The highest LS between anthranilate synthase A and anthranilate synthase B1, as well as tryptophan synthase A and tryptophan synthase B1 clearly shows that these subunits are the preferred subunits for the assembly of anthranilate synthase and tryptophan synthase complex respectively.

## Figures and Tables

**Figure 1 plants-12-01687-f001:**
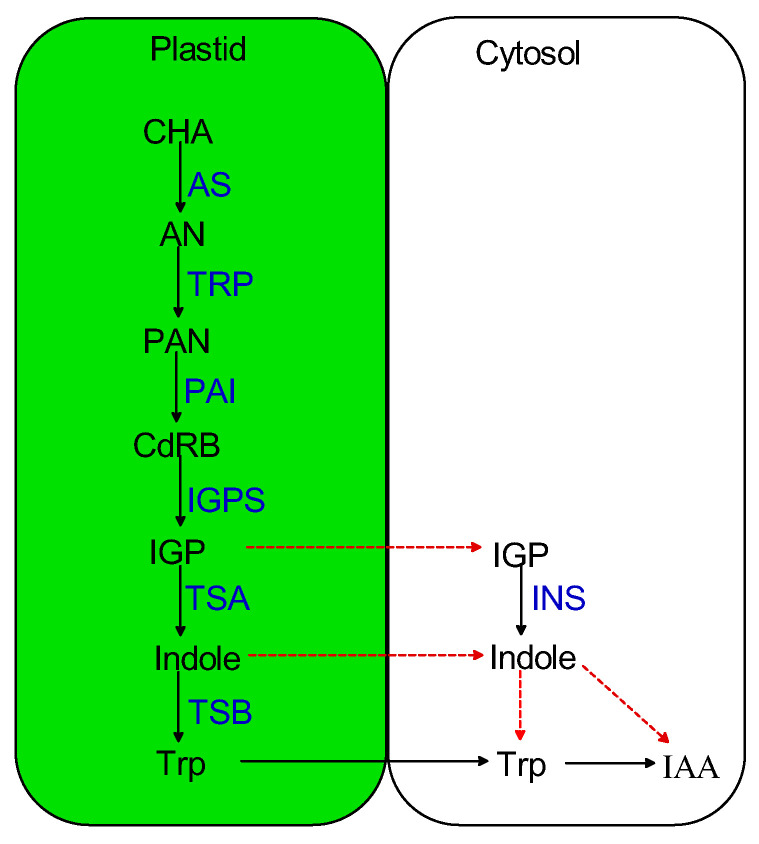
The plastidal and cytosolic pathways of indole production in plants. CHA, chorismate; AS, anthranilate synthase; AN, anthranilate; TRP, phosphoribosyl anthranilate transferase; PAN, 5-phosphoribosylanthranilate; PAI, phosphoribosyl anthranilate isomerase; CdRB; 1-(o-carboxyphenylamino)-1-deoxy-ribulose 5-phosphate; IGPS, indole-3-glycerol phosphate synthase; IGP, indole-3-glycerol phosphate; TSA, tryptophan synthase α subunit; TSB, tryptophan synthase β subunit; INS, indole synthase. Dotted lines denote unconfirmed reactions.

**Table 1 plants-12-01687-t001:** List of genes reported to have an important role in the indole and trp synthesis. ASA, anthranilate synthase A; ASB, anthranilate synthase B; TRP, phosphoribosyl anthranilate transferase; PAI, phosphoribosyl anthranilate isomerase; IGPS, indole-3-glycerol phosphate synthase; TSA, tryptophan synthase A; TSB, tryptophan synthase B, INS; indole synthase.

Gene	AGI Locus	Gene	AGI Locus
*ASA1*	At5g05730	*PAI3*	At1g29410
*ASA2*	At2g29690	*IGPS*	At2g04400; At5g48220
*ASA3*	At3g55870	*TSA*	At3g54640
*ASB1*	At1g25220	*TSB1*	At5g54810
*ASB2*	At5g57890	*TSB2*	At4g27070
*ASB (putative)*	At1g24807 At1g24909 At1g25083 At1g25155	*TSB3*	At5g38530
*TRP1*	At5g17990; At1g70570	*TSB4*	At5g28237
*PAI1*	At1g07780	*INS*	At4g02610
*PAI2*	At5g05590		

**Table 2 plants-12-01687-t002:** Coexpression of *TSA* with *TSB1*, *TSB2*, *TSB3*, and *TSB4* under general and specific conditions. Coexpression is supported by the latest release of RNA sequencing (Ath-r.5) and microarray data (Ath-m.9). Logit Score (LS) was used as a coexpression index. Higher LS indicates a better coexpression performance. *TSA*; *tryptophan synthase A*, *TSB*; *tryptophan synthase B*. Specific conditions include: tissue, a biotic, hormone, biotic, and light.

Gene	AGI Locus	*TSA* At3g54640	Platform
*TSB1*	At5g54810		Ath-u.2 ^2^
		10.8	Ath-r.5
*TSB2*	At4g27070	6.4	Ath-u.2
		6.8	Ath-r.5
*TSB3*	At5g38530	2.1 ^1^	Ath-u.2
		1.6 ^1^	Ath-r.5
		2.1 ^1^	Ath-m.9
		2.7 ^1^	Abiotic
		3.6	Hormone

^1^ Nonsignificant coexpression index. ^2^ Ath-u.2, unified coexpression platform.

**Table 3 plants-12-01687-t003:** Coexpression of *TSA* with trp biosynthesis genes in the chorismic acid pathway under general and specific conditions. Coexpression is supported by the latest release of RNA sequencing (Ath-r.5) and microarray data (Ath-m.9). Logit Score (LS) was used as a coexpression index. Higher LS indicates a better coexpression performance. *ASA*; *anthranilate synthase A*, *ASB*; *anthranilate synthase B*, *TRP*; *phosphoribosyl anthranilate transferase*, *PAI*; *phosphoribosyl anthranilate isomerase*, *IGPS*; *indole-3-glycerol phosphate synthase*. Specific conditions include: tissue, a biotic, hormone, biotic, and light. Ath-u.2; unified coexpression platform.

Gene	Ath-u.2	Ath-r.5	Ath-m.9	Tissue	Stress	Hormone	Biotic	Light
*ASA1*	12.9	11.6	10.5	5.0	4.4	4.3	5.3	4.8
*ASB1*	7.9	8.2	-	-	-	-	-	-
*TRP1*	12.9	11.8	10.3	3.8	5	5.2	5.2	4.4
*IGPS1*	17.3	13.8	16.0	4.0	5.7	5.7	5.6	-

**Table 4 plants-12-01687-t004:** Coexpression of *IGPS1*, *ASA1*, *TRP1*, and *ASB1* as a bait set under general conditions in the ath-u.c2-0 platform. The new ranks of genes are calculated in respect to the query genes. Logit Score (LS) was used as a coexpression index. Higher LS indicates a better coexpression performance. *IGPS*; *indole-3-glycerol phosphate synthase*, *ASA*; *anthranilate synthase A*, *TRP*; *phosphoribosyl anthranilate transferase*, *ASB*; *anthranilate synthase B*, *TSA*; *tryptophan synthase A*, *TSB*; *tryptophan synthase B*.

Gene	AGI Locus	Average LS to Query Genes	New Rank
* **IGPS1** *	**At2g04400**	**8.06**	**6**
* **ASA1** *	**At5g05730**	**7.48**	**10**
* **TRP1** *	**At5g17990**	**7.32**	**13**
* **ASB1** *	**At1g25220**	**5.4**	**30**
*TSA1*	At3g54640	12.73	1
*TSB1*	At5g54810	7.52	8
*TSB2*	AT4G27070	5.03	33

## Data Availability

Not applicable.

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
