# Peer review of "Comparative Coexpression Analysis of Indole Synthase and Tryptophan Synthase A Reveals the Independent Production of Auxin via the Cytosolic Free Indole"

_plants, 2023, doi:10.3390/plants12081687_

Round 1
Reviewer 1 Report
Indole synthase (INS) was reported as the first enzyme in the tryptophan-independent pathway of auxin synthesis. The main aim of this study was to find out whether INS is involved in the tryptophan-dependent or independent pathway and demonstrated some important results.
In Geneeral, this manuscript is well-written and the data analyses are good and results provide some new information about the role of INS involved in the tryptophan-dependent or independent pathway. I have only following minor points:
1. The end of the abstract should the final conclusion for this study.
2. In the introduction, there is no reference from 2023.
3. All the figures are not in the balance. Figure 1, Figure narrow, figure title too wide
4. Table 1-4 have the same problem. The Tables ares much wider than the Table titles are narrow. 、
5. References are not consistent, some have DOI number, some not, please make them all the same.
Author Response
- The end of the abstract should the final conclusion for this study. The last 7 sentences are all conclusion “However, INS was not found to coexpressed with any target genes suggesting that it may exclusively and independently involves in the tryptophan-independent pathway. Additionally, annotation of examined genes as ubiquitous or differentially expressed were described and subunits-encoded genes available for the assembly of tryptophan and anthranilate synthase complex were suggested. The most probable TSB subunits expected to interact with TSA is TSB1 then TSB2. Whereas TSB3 is only used under limited hormone conditions to assemble tryptophan synthase complex, putative TSB4 is not expected to be involved in the plastidial synthesis of tryptophan in Arabidopsis.”
- In the introduction, there is no reference from 2023. Three new references dated in 2023 were added to the introduction and highlighted in red.
- All the figures are not in the balance. Figure 1, Figure narrow,figure title too wide
- Table 1-4 have the same problem. The Tables ares much wider than the Table titles are narrow. I think this problem was already resolved by the journal format.
- References are not consistent, some have DOI number, some not, please make them all the same. DOI for all references were inserted and highlighted in red.
Reviewer 2 Report
In the manuscript entitled “Comparative coexpression analysis of INS and TSA reveals the independent production of auxin via the cytosolic free indole”, the authors performed coexpression analysis using the ATTED-II data base and showed that Indole synthase (INS) is not coexpressed with genes in the tryptophan dependent pathway of auxin biosynthesis, thereby suggesting that INS may function exclusively in tryptophan independent pathways of auxin biosynthesis. The cytosolic enzyme INS which converts indole-3-glycerol phosphate to indole is homologous to the plastidal tryptophan synthase A (TSA) enzyme. However, while previous work indicated that INS is the first enzyme in the tryptophan-independent pathway of auxin biosynthesis, one study proposed that INS or its indole product may interact with TSB4 and therefore may be involved in the tryptophan dependent pathway of auxin biosynthesis. Therefore, the goal of this manuscript was to use coexpression analysis to determine the involvement of INS in the tryptophan dependent or tryptophan independent pathway of auxin biosynthesis. The coexpression results show that INS does not coexpress with TSB genes, suggesting that it functions in auxin and indole biosynthesis independently of the tryptophan pathway.
Overall, this manuscript is well written. The authors’ coexpression results support their conclusions and the analysis of the data are adequate for the overall objective of the study. The tables are clear, easy to read and to understand.
A few minor suggestions:
· Perhaps the authors should write the entire name for indole synthase and tryptophan synthase A in the title of the manuscript instead of the abbreviations.
· The font and size of the text in the manuscript is inconsistent. The authors should make these consistent throughout the manuscript.
Author Response
- Perhaps the authors should write the entire name for indole synthase and tryptophan synthase A in the title of the manuscript instead of the abbreviations. Changes have been done and highlighted in red
- The font and size of the text in the manuscript is inconsistent. The authors should make these consistent throughout the manuscript. Resolved by the new format of the journal